# Realizing topological edge states with Rydberg-atom synthetic dimensions

S. K. Kanungo [1,2 ✉], J. D. Whalen[1,2], Y. Lu[1,2], M. Yuan [1,2,3,4], S. Dasgupta[1,2], F. B. Dunning[1],
K. R. A. Hazzard[1,2 ✉] & T. C. Killian [1,2 ✉]

A discrete degree of freedom can be engineered to match the Hamiltonian of particles moving in a real-space lattice potential. Such synthetic dimensions are powerful tools for quantum simulation because of the control they offer and the ability to create configurations difficult to access in real space. Here, in an ultracold $^{84}$Sr atom, we demonstrate a synthetic-dimension based on Rydberg levels coupled with millimeter waves. Tunneling amplitudes between synthetic lattice sites and on-site potentials are set by the millimeter-wave amplitudes and detunings respectively. Alternating weak and strong tunneling in a one-dimensional configuration realizes the single-particle Su-Schrieffer-Heeger (SSH) Hamiltonian, a paradigmatic model of topological matter. Band structure is probed through optical excitation from the ground state to Rydberg levels, revealing symmetry-protected topological edge states at zero energy. Edge-state energies are robust to perturbations of tunneling-rates that preserve chiral symmetry, but can be shifted by the introduction of on-site potentials.

[1] Department of Physics and Astronomy, Rice University, Houston, TX 77005-1892, USA. [2] Rice Center for Quantum Materials, Rice University, Houston, TX 77005-1892, USA. [3] School of the Gifted Young, University of Science and Technology of China, Hefei 230026, China. [4] Present address: Pritzker School of Molecular Engineering, University of Chicago, Chicago, IL 60637, USA. ✉email: skk4@rice.edu; kaden@rice.edu; killian@rice.edu

A synthetic dimension[1,2] is a degree of freedom encoded into a set of internal or external states that can mimic motion of a particle in a real-space lattice potential. Synthetic dimensions are powerful tools for quantum simulation, opening exciting possibilities such as the realization of higher dimensional systems[1,3,4], non-trivial real space[5,6] and band structure[7,8] topologies, and artificial gauge fields[9,10]. These can be used in conjunction with real space lattices to create situations unavailable to either method individually. Experiments have utilized various degrees of freedom[2] to create synthetic dimensions, such as motional[11,12], spin[9,13–15], and rotational[16] levels of atoms and molecules, and frequency modes, spatial modes, and arrival times in photonic systems[2].

Prominent demonstrations of atomic synthetic dimensions include observation of artificial gauge fields, spin-orbit coupling, and chiral edge states in Raman-coupled ground magnetic sublevels[9,13,17] or single-photon-coupled electronic orbitals[18,19] grafted onto motion in a real 1D optical lattice. A synthetic dimension can also be formed by discrete motional states[20], such as free-particle momentum states coupled with momentum-changing two-photon Bragg transitions[21,22]. The latter has been used to observe Anderson localization[23], artificial gauge fields[24], and topological states[25,26].

Here we harness Rydberg levels of $^{84}$Sr to realize a synthetic lattice for studying quantum matter. Rydberg levels $|i\rangle$ and $|j\rangle$ coupled with amplitude $\Omega_{ij}$ by resonant millimeter waves are described by the same Hamiltonian as a particle tunneling between lattice sites $|i\rangle$ and $|j\rangle$ with tunneling amplitude $J_{ij} = \Omega_{ij}/2$. Because of this mathematical equivalence to particles moving in a real-space lattice, coupled Rydberg levels can function as a synthetic spatial dimension. This scheme, first suggested in[2], is similar to a proposal for synthetic dimensions based on molecular rotational levels[27–29]. It allows for control of the connectivity, tunneling rates, and on-site potentials, and creation of a broad range of synthetic dimensional systems, including systems not realizable in physical space. The number of available Rydberg levels and strong transition dipole moments make large and complex synthetic landscapes feasible. Rydberg dipole-dipole

interactions[30] provide a mechanism for creating tunable, localized interactions for many-body systems in synthetic space, which is a challenge for other atom-based platforms. The concept of a synthetic dimension was recently used to explain conical intersections in the potential energy curves of Rydberg molecules[31].

To demonstrate the capabilities of Rydberg-atom synthetic dimensions, we realize the Su-Schrieffer-Heeger (SSH) model[32] in synthetic space (Fig. 1), and study its topologically protected edge states (TPS) and their robustness to disorder. The SSH model describes a linear conjugated polymer, such as polyacetylene, with alternating weak and strong tunneling. The configuration with weak tunneling to edge sites possesses doubly degenerate TPS with energy centered in the gap between bulk states [Fig. 1d]. TPS energies are robust against perturbations respecting the chiral symmetry of the tunneling pattern[33,34], as observed in many systems[25,35–37].

## Results

**Creating and Probing the Synthetic Lattice.** The essential elements of the apparatus are shown in Fig. 1a. $^{84}$Sr atoms are trapped in an optical dipole trap at a peak density of about $10^{11}$ cm$^{-3}$ and a temperature of $T = 2\,\mu$K. Millimeter waves are switched on to provide couplings as shown in Fig. 1b or c and construct a six-site synthetic lattice with three $5sns\ ^3S_1(m=1)$ ($\equiv ns$, sites $i = 1, 3, 5$, with $57s$ mapped to $i = 1$) and three $5snp\ ^3P_0$ ($\equiv np$, $i = 2, 4, 6$) levels. The resulting Hamiltonian is

$$\hat{H}_{\text{lattice}} = \sum_{i=1}^{5}(-hJ_{i,i+1}|i\rangle\langle i+1| + \text{h.c.}) + \sum_{i=1}^{6}h\delta_i|i\rangle\langle i|, \quad (1)$$

where $J_{i,i+1}$ are the tunneling amplitudes and $\delta_i$ are on-site potentials set respectively by amplitudes and detunings of the millimeter-wave couplings, and $h$ is Planck's constant. To obtain Eq. (1), we have neglected counter-rotating terms in the millimeter-wave couplings and transformed into a rotating frame. The kets $|i\rangle$ correspond to the unperturbed Rydberg levels of $^{84}$Sr up to a time-dependent phase arising from the transformation.

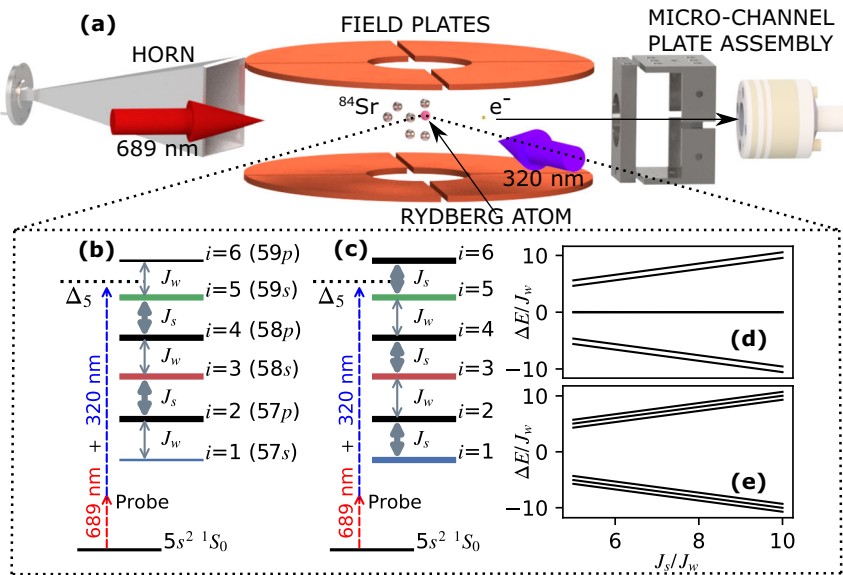

**Fig. 1 Implementing the six-site Su-Schrieffer-Heeger (SSH) model.** (**a**) Experimental schematic. (**b**) and (**c**) represent realizations of the SSH model using six Rydberg levels of $^{84}$Sr that, respectively do and do not possess topologically protected edge states (TPS). Double-headed gray arrows denote near-resonant millimeter-wave couplings, which induce tunneling between sites of the synthetic lattice, and thicker lines correspond to faster tunneling. Dashed lines show two-photon excitation to a Rydberg level of interest. (**d**, **e**) show band structure for (**b**) and (**c**) respectively vs. the ratio of strong and weak tunneling amplitudes, $J_s/J_w$. The site-numbering convention is given in (**b**), with odd numbers corresponding to $ns$ states.

$\delta_i = 0$ yields the SSH model, and the configuration with TPS has $J_{i,i+1} = J_w$ ($J_s$) for $i = 1, 3, 5 (2, 4)$ and $J_w < J_s$. For the configuration without TPS, the weak and strong couplings are exchanged. Here, the strong coupling $J_s$ is varied from 0.5–1.5 MHz, and all data is taken with weak coupling $J_w = 100$ kHz ($J_s/J_w = 5$–15). A 4 Gauss magnetic field creates Zeeman splittings that ensure millimeter-wave couplings to $5sns\ ^3S_1(m = -1, 0)$ states are negligible.

To populate and probe the synthetic space, the $^{84}$Sr ground state is coupled to the Rydberg levels via two-photon excitation using the intermediate $5s5p\ ^3P_1$ level[38–40], applied in a 5 $\mu$s pulse. The laser polarizations select excitation to $ns(m = 1)$ levels. Immediately after excitation, Rydberg populations are detected using selective field ionization (SFI)[41], in which an electric-field ramp liberates any electrons in an excited Rydberg state. The electron arrival time on a charged-particle detector heralds the Rydberg level, or occupied synthetic-lattice site. With the current experimental resolution, arrival times for states $np$ and $(n + 1)s$ are unresolved.

To probe the lattice band structure, the two-photon excitation laser is tuned, with detuning $\Delta_{i_{\mathrm{pr}}}$, near the energy of one of the unperturbed Rydberg levels $|i_{\mathrm{pr}}\rangle$ [Fig. 1b, c]. Neglecting far off-resonant terms, the Hamiltonian for the entire system can be written as:

$$\hat{H} = \frac{h\Omega_{i_{\mathrm{pr}}}}{2} |g\rangle\langle i_{\mathrm{pr}}| e^{i2\pi\Delta_{i_{\mathrm{pr}}}t} + \mathrm{h.c.} + \hat{H}_{\mathrm{lattice}}, \qquad (2)$$

where $\Omega_{i_{\mathrm{pr}}}$ denotes the effective two-photon Rabi frequency, which vanishes for even $i_{\mathrm{pr}}$ ($np$ levels), and $|g\rangle$ is the ground state vector in the frame rotating at the frequency difference of the $|i_{\mathrm{pr}}\rangle$ and $|g\rangle$ levels. The Rydberg excitation rate before convolving with instrumental linewidth is well-described as

$$\Gamma(\Delta_{i_{\mathrm{pr}}}) = \pi^2 \Omega_{i_{\mathrm{pr}}}^2 \sum_\beta |\langle\beta|i_{\mathrm{pr}}\rangle|^2 \delta(\Delta_{i_{\mathrm{pr}}} - \epsilon_\beta/h), \qquad (3)$$

where $|\beta\rangle$ and and $\epsilon_\beta$ are the eigenstates and eigenenergies of $\hat{H}_{\mathrm{lattice}}$ [Fig. 1d, e]. These eigenstates in the synthetic dimension may alternatively be viewed as atomic states dressed by the millimeter-wave field. Indeed, strong photon coupling of atomic levels is often described in the language of Autler-Townes splitting[42,43], and the coupling Rabi frequency is related to the tunneling rate through $\Omega = 2J$. But with increasing system size, the lattice interpretation becomes more natural: a band structure emerges even for six levels demonstrated here, as do phenomena, such as edge states with an energy splitting that is exponentially small in the number of levels.

## SSH Band Structure and State Decomposition.
A collection of spectra, with each spectrum arising from coupling the ground state $|g\rangle$ to a different lattice site $|i_{\mathrm{pr}}\rangle$, complement each other to provide a characterization of the band structure and decomposition of the eigenstates because the spectral contribution from each eigenstate is proportional to its overlap with the unperturbed Rydberg level corresponding to the lattice site $i_{\mathrm{pr}}$.

Figure 2a shows spectra for the configuration with TPS, $J_s/J_w = 5$, and $\delta_i = 0$ as a function of probe-laser detuning near each of the unperturbed Rydberg $ns$ levels (odd $i_{\mathrm{pr}}$). Each spectrum is normalized by the total signal for its $i_{\mathrm{pr}}$. Contributions to the spectra near $\Delta_{i_{\mathrm{pr}}} = 0$ (edge states) correspond to population localized at the upper and lower boundaries of the lattice. Edge states in a gap of width $\sim 2J_s$ between bulk states are hallmark features of the SSH model.

The edge-state signal is large for probe detuning near the $57s$ level ($i_{\mathrm{pr}} = 1$), small for the $58s$ ($i_{\mathrm{pr}} = 3$) spectrum, and barely

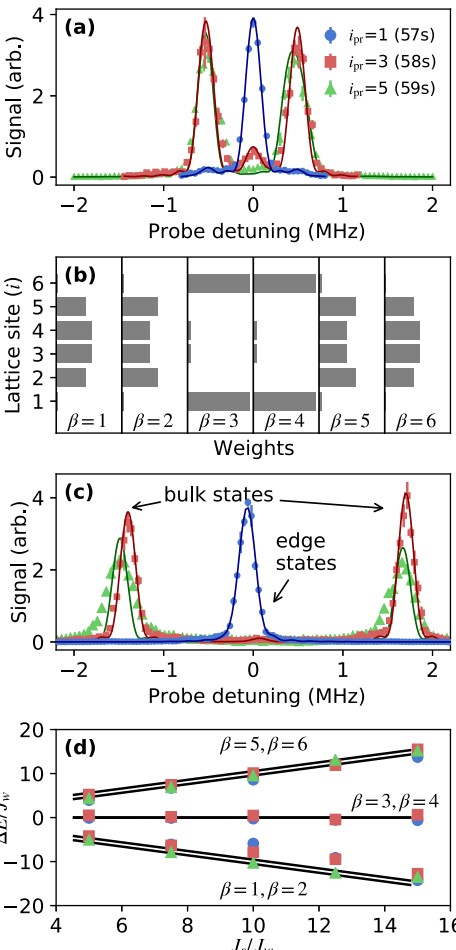

**Fig. 2 Band structure and state decomposition for the configuration with TPS [Fig. 1b].** (**a**) Rydberg excitation spectra when coupling to $i_{\mathrm{pr}} = 1(57s)$, $i_{\mathrm{pr}} = 3(58s)$, and $i_{\mathrm{pr}} = 5(59s)$ for $J_s/J_w = 5$. Probe detuning ($\Delta_{i_{\mathrm{pr}}}$) is from the undressed Rydberg level. $i_{\mathrm{pr}} = 3, 5$ spectra have been multiplied by a factor of 2 for clarity. (**b**) State decomposition weights, $|\langle\beta|i\rangle|^2$, obtained from direct diagonalization of Eq. (1) for $J_s/J_w = 5$. Edge states correspond to $\beta = 3, 4$. (**c**) Same as (**a**) except for $J_s/J_w = 15$. (**d**) Peak positions ($\epsilon_\beta$) in spectra such as in (**a**, **c**), giving the bulk and edge state energies versus $J_s$. Measurements match the band structure calculated by direct diagonalization of Eq. (1). Bulk states split by $\sim J_w$ are unresolved in the spectra. All energies are scaled by $J_w$. Lines in (**a**) and (**c**) are fits of each feature with a sinc-squared lineshape corresponding to the 5 $\mu$s laser exposure time convolved with a 100 kHz full-width-half-maximum (FWHM) additional broadening. All error bars denote one standard error (s.e.) of the mean.

observable for $59s$ ($i_{\mathrm{pr}} = 5$). The integrated signal intensity around the peak centered at detuning $\Delta_{i_{\mathrm{pr}}} = \epsilon_\beta/h$ reflects the overlap of the lattice eigenstate $|\beta\rangle$ with $|i_{\mathrm{pr}}\rangle$ [Eq. (3)]. Thus, the intensity pattern confirms that the edge states are localized on the weakly coupled boundary sites, with little contribution from undressed bulk sites $58s$ ($i = 3$) and $59s$ ($i = 5$). This matches the expected decomposition of each SSH eigenstate $|\beta\rangle$ upon the bare lattice sites, expressed in the factors $|\langle\beta|i\rangle|^2$, which can be obtained from direct diagonalization of Eq. (1) [Fig. 2b, $\beta = 3, 4$ correspond to edge states]. The widely split bulk states, however, give rise to the approximately equal spectral contributions at $\Delta_{i_{\mathrm{pr}}} \approx \pm J_s$ in Fig. 2a, revealing the energy splitting in the band structure. The bulk-state features are strong for gross detuning near the $58s$ and $59s$ undressed levels, and very weak near $57s$,

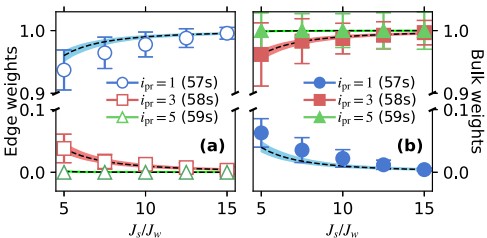

**Fig. 3 Synthetic-lattice-eigenstate decomposition obtained from spectral-line areas [e.g. Fig. 2a, c] for the configuration with TPS [Fig. 1b].** Error bars reflect fit uncertainties. (**a**) Fraction of the entire signal under the spectral features corresponding to the edge states for probe tuned near site $i_{pr}$ (Rydberg level) indicated in the legend. The line is the sum of the squares of the calculated overlaps of the SSH edge eigenstates with the $i_{pr}$ site found from a direct diagonalization of Eq. (1) with $\delta_i = 0$ [e.g. Fig. 2b]. (**b**) Fraction of the entire signal under the bulk state features and calculated sum of squares of the overlaps of the SSH bulk eigenstates with the $i_{pr}$ site.

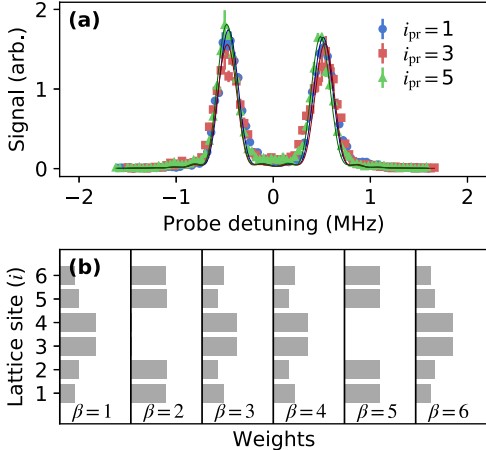

**Fig. 4 Study of coupling configuration without TPS [Fig. 1c] at $J_s/J_w = 5$.** (**a**) Rydberg excitation spectra when coupling to $i_{pr} = 1(57s)$, $i_{pr} = 3(58s)$, and $i_{pr} = 5(59s)$. Probe detuning ($\Delta_{i_{pr}}$) is from the undressed Rydberg level. Lines are the same as in Fig. 2. Error bars denote one s.e. of the mean. (**b**) State decomposition weights, $|\langle \beta | i \rangle|^2$, obtained from direct diagonalization of Eq. (1). All states have appreciable bulk character.

which is expected because the bulk-state decompositions [$\beta = 1, 2, 5, 6$ in Fig. 2b] show little weight on edge sites $57s$ ($i = 1$) and $59p$ ($i = 6$).

Figure 2c shows spectra for stronger coupling $J_s/J_w = 15$, also in the configuration with TPS. The edge state contributions at $\Delta_{i_{pr}} = 0$, indicate greater localization to $57s$ ($i = 1$) than for $J_s/J_w = 5$. Splitting for the bulk states matches $\Delta_{i_{pr}} \approx \pm J_s$. From the peak positions in a series of data sets such as Fig. 2a, c, the band structure as a function of strong-tunneling rate $J_s$ can be measured [Fig. 2d]. It agrees with results from a direct diagonalization of Eq. (1) with $\delta_i = 0$.

A series of data sets such as Fig. 2a, c can also be used to study the variation in state decomposition as a function of strong-tunneling rate $J_s$. Exact diagonalization, such as Fig. 2b, provides the decomposition of each SSH eigenstate $|\beta\rangle$ upon the bare lattice sites, expressed in the factors $|\langle\beta|i\rangle|^2$. This can be compared with experimental measurements of the fraction of the total spectral area in either the edge or the bulk spectral features when probing the overlap with a specific lattice site ($i_{pr}$) in spectra such as Fig. 2a, c. Spectral area is determined by fitting

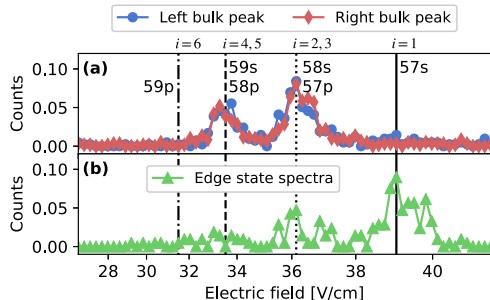

**Fig. 5 Detection of edge and bulk states by selective field ionization (SFI).** SFI signals for probe laser tuned near the $58s$ Rydberg level ($i_{pr} = 3$) for $J_s/J_w = 5$ [Fig. 2a]. Vertical lines indicate ionization fields for bare Rydberg levels. Data points are evenly spaced in time. (**a**) For excitation to the left and right bulk-state peaks ($\Delta_{i_{pr}} \approx \pm 500$ kHz), the state excited is localized on the bulk sites of the synthetic lattice. ((**b**) For excitation to the edge-state peak, ($\Delta_{i_{pr}} \approx 0$) the state is localized on the $57s$ ($i = 1$) boundary site. The small contribution to the signal at $i = 2, 3, 4$, and 5 is predominantly from the wings of the bulk-state peaks.

each of the three features in a spectrum with a sinc-squared lineshape corresponding to the $5\,\mu s$ laser exposure time convolved with a 100 kHz FWHM Gaussian linewidth from laser and natural broadening, estimated from independent measurements of spectra in the absence of millimeter-wave fields. Center frequency and amplitude are varied for fitting. Figure 3 (left) shows that the experimentally measured edge-state fraction matches $\sum_{\beta \in \text{edge}} |\langle \beta | i_{pr} \rangle|^2$, and Fig. 3 (right) does the same for the bulk contribution and $\sum_{\beta \in \text{bulk}} |\langle \beta | i_{pr} \rangle|^2$. The width of the calculated line denotes 10% variation in the Rabi frequencies. For a given $J_s/J_w$, the edge-state measurements in Fig. 3 add to one, while the bulk-state measurements add to two. This reflects the fact that there are two edge states and four bulk states for this system, and half of the weight for the states in each group is in overlap with even lattice sites, which the photoexcitation probe does not detect.

For the configuration with strong tunneling to the boundary sites, which should not have TPS, the Rydberg excitation spectra show unresolved bulk states split by $\Delta_{i_{pr}} \approx \pm J_s$, with no states in the gap between them [Fig. 4a]. A schematic of the full band structure is shown in Fig. 1e. The state decomposition from direct diagonalization of Eq. (1) shows that all states have appreciable bulk character and there are no edge states [Fig. 4b].

**State Decomposition with Selective Field Ionization.** Because the spectral probe is only sensitive to $ns$ contributions to the state vector (odd $i$), it cannot establish whether the edge states observed are localized on one boundary site or a superposition of both. To answer that question, we turn to SFI as a tool for site-population measurements in Rydberg-atom synthetic dimensions. For Rydberg excitation near $58s$, corresponding to $i_{pr} = 3$, and for $J_s/J_w = 5$ [Fig. 2a], if the detuning is set to resonance with the left or right bulk-state peaks ($\Delta_{i_{pr}} \approx \pm 500$ kHz), electrons are liberated at ionization fields for Rydberg levels corresponding entirely to bulk sites of the synthetic lattice ($i = 2-5$) [Fig. 5a]. For laser detuning on the edge-state peak ($\Delta_{i_{pr}} \approx 0$), signal arrives at fields corresponding predominantly to the $57s$ Rydberg state ($i = 1$) [Fig. 5b]. This indicates localization of the edge state on the boundary in general and, more specifically, on the single boundary site connected to the ground-state by the two-photon excitation [Eq. (3)], which is a linear combination of states $\beta = 3$ and $\beta = 4$. Integrals of SFI signals corresponding to each lattice

site and for each spectral feature provide a state decomposition that agrees with expectations as in Fig. 3.

**Protected Edge States.** The pinning of the edge-state energy to $\Delta_{i_{pr}} = 0$ is the defining feature of TPS in the SSH model. It arises because of an underlying chiral symmetry, which reflects the system's bipartite structure (even and odd sites) with all Hamiltonian matrix elements vanishing between sites of the same partition, including diagonal (on-site) matrix elements. To investigate the robustness of the pinning of the edge-state energy, we probe the band structure in the presence of perturbations from the SSH form.

Figure 6a–c shows spectra for $i_{pr} = 3$ (58s) and $i_{pr} = 5$ (59s) for balanced ($J_{2-3} = J_{4-5} = J_s^0$) and imbalanced [$J_{2-3} = (1 \pm 0.15)J_s^0$ and $J_{4-5} = (1 \mp 0.15)J_s^0$] strong coupling with $J_s^0/J_w = 5$. The bulk states are strongly affected by imbalance. With increased $J_{2-3}$ [Fig. 6b], the two bulk states that are more localized on the $i = 3$ site show increased splitting. With increased $J_{4-5}$ [Fig. 6c], the two bulk states that are more localized on the $i = 5$ site show increased splitting. The energy of the edge-state signal, however [in Fig. 6b, c], is immune to this perturbation, which preserves the protecting chiral symmetry because the tunneling matrix elements only connect even and odd sites.

Figure 6d, e shows how the energies of the edge states are affected by chiral-symmetry-breaking perturbations, in particular shifts of on-site potentials (i.e. millimeter-wave coupling frequencies). Spectra are recorded with the probe laser tuned near the 58s level ($i_{pr} = 3$) for $J_s/J_w = 10$. For Fig. 6d, the frequency of the $i = 1$ to $i = 2$ (57s-57p) coupling is varied, which shifts $\delta_1$, the on-site potential of the $i = 1$ (57s) site in the synthetic lattice. $\delta_1 \neq 0$ yields a diagonal term in the Hamiltonian [Eq. (1)] that breaks the chiral symmetry, and the edge-state

signal shifts by an amount equal to the detuning from resonance. For Fig. 6e, the frequency of the $i = 5$ to $i = 6$ (59s-59p) coupling is varied, shifting $\delta_6$, and the position of the edge-state signal remains unchanged. These results confirm that the edge state coupled to by the probe laser is localized on the $i = 1$ (57s) boundary site. The orthogonal edge state is localized on $i = 6$, with vanishing weight on odd sites. In general we expect that any perturbation producing a Hamiltonian term that connects only even sites to even sites, or odd to odd, will break the chiral symmetry and shift edge-state energies. This particular form of perturbation only affects the energy of one of the edge states.

**Numerical Simulations of the Spectra and Effects of Decoherence.** In order to gain more insight into the system and explore the effects of decoherence, we perform theoretical calculations based on the Lindblad master equation [Eq. (5), given in the Methods] for the Hamiltonian given in Eq. (2). We consider one model with no decoherence and one model with decoherence in the form of white amplitude noise on the millimeter waves. The choice of decoherence model, and other, less important, modifications of the idealized picture, are discussed in the Methods. The master equation is derived from stochastic Schrödinger equations following standard arguments[44]. Results are convolved with a 100 kHz FWHM additional broadening for the probe field. The dominant experimental source of decoherence is not yet determined, but amplitude noise is a simple model consistent with observations. Moreover, the qualitative conclusions we reach are not sensitive to the choice of noise, and would also hold if the noise source were, for example, fluctuating magnetic fields or frequency noise on the millimeter waves.

The theory with no decoherence fits the spectra extremely well (Figs. 7 and 8), in particular reproducing the intensities and linewidths of the observed spectral features, but with a few notable exceptions as discussed below. In the calculations shown in the figures, we allow the $J_{i,i+1}^{i_{pr}}$ and the $\delta_i^{i_{pr}}$ to vary from their nominal values for the strong bonds, fitting them for each spectrum (i.e. for each value of $i_{pr}$). The values determined from fitting agree with measured values within experimental uncertainties. Linewidths from the numerical simulation match the time-broadened widths of the sinc-squared lineshapes used in Figs. 2 and 4 to extract spectral areas for comparison with direct diagonalizaion of Eq. (1) (Fig. 3).

Discrepancies between observations and decoherence-free theory are visible in the linewidths for $i_{pr} = 5$ for the largest value of strong coupling $J_s$ (Fig. 7). Other differences between theory and experiment are the small reductions in contrast between the spectral features and a very small increase in weight and smoothing in the tails of the spectra, which are seen most strongly for $i_{pr} = 5$ (Fig. 8). Calculations with decoherence

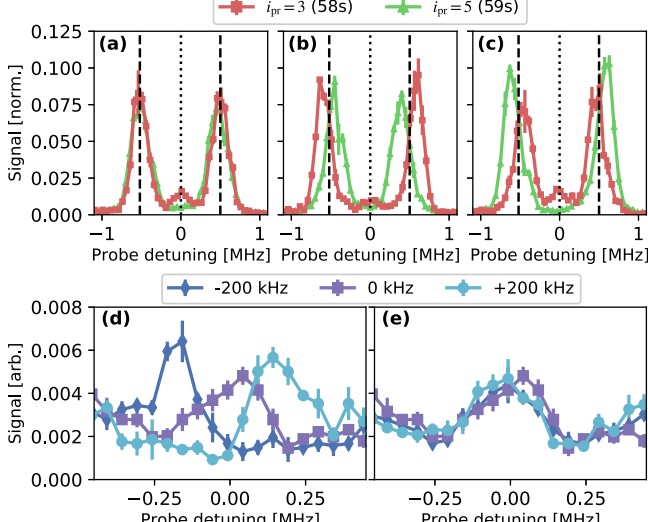

**Fig. 6 Band structure with Hamiltonian perturbations.** (**a**) SSH model: $\delta_i = 0$ with balanced tunneling rates ($J_{2-3}$, $J_{4-5} = J_s^0$ and $J_s^0/J_w = 5$). Lines mark positions of the bulk and edge peaks. (**b**) Strong tunneling rates are imbalanced to $J_{2-3} = 1.15J_s^0$ and $J_{4-5} = 0.85J_s^0$ for $J_s^0/J_w = 5$. This perturbation respects chiral symmetry. (**c**) Same as (**b**) but $J_{2-3} = 0.85J_s^0$ and $J_{4-5} = 1.15J_s^0$. (**d**) Edge state spectra in the presence of perturbations breaking chiral symmetry: tunneling rates are balanced as in the standard topological SSH configuration, with $J_s/J_w = 10$, but the frequency of the $i = 1$ to $i = 2$ (57s-57p) coupling is varied by the value given in the legend. Spectra are recorded with the probe laser tuned near the 58s level ($i_{pr} = 3$). (**e**) Same as (**d**), but the $i = 5$ to $i = 6$ (59s-59p) coupling frequency is varied. All error bars denote one s.e. of the mean.

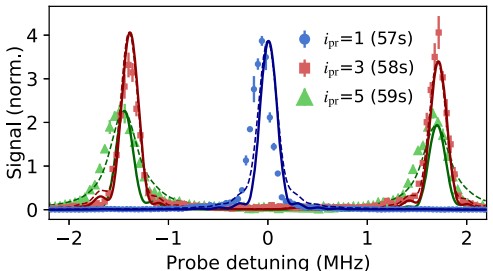

**Fig. 7 Noise model analysis.** Fits of experimental data with theoretical calculations assuming no decoherence (———), and using the amplitude-noise model [Eq. (6), – – – –]. Data shown are for $J_s/J_w = 15$ [from Fig. 2c, the $i_{pr} = 3, 5$ spectra have been multiplied by a factor of 2 for clarity]. All error bars denote one s.e. of the mean.

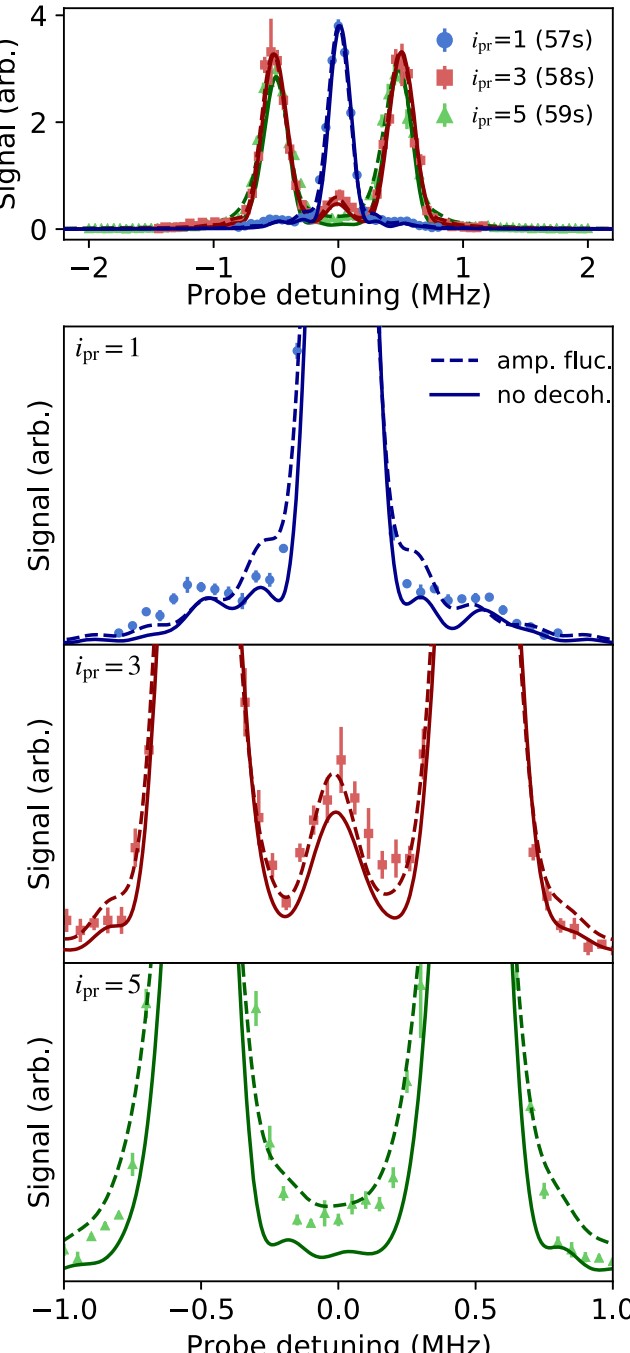

**Fig. 8 Noise model analysis.** Same as Fig. 7, but experimental data shown are for $J_s/J_w = 5$ [from Fig. 2a]. Top panel shows all three $i_{pr}$ while the bottom three panels show central regions of the same data for the individual $i_{pr}$ indicated in each panel. All error bars denote one s.e. of the mean.

capture these effects and provide values for decoherence rates $\Gamma_i^{i_{pr}}$ by fitting to the spectra for each value of $i_{pr}$. For simplicity, we assume that $\Gamma_i^{i_{pr}} = \mathcal{C}^{i_{pr}} J_{i,i+1}$, which constrains $\Gamma_i^{i_{pr}}$ to be proportional to the microwave amplitude for the associated bond, with the single proportionality constant ($\mathcal{C}^{i_{pr}}$) determined by fitting.

Trends in the decoherence emerge from this analysis. One already mentioned is that the decoherence for $i_{pr} = 5$ is much larger than for other $i_{pr}$ values when in the topological tunneling configuration, which is reflected in a value of $\mathcal{C}^{i_{pr}=5}$ that is $\sim 3$

times larger than for other $i_{pr}$. When in the trivial configuration, no such difference is observed. A possible explanation is that the fluctuations on the coupling between $|5\rangle$ and $|4\rangle$ are particularly noisy. Another possibility is coupling of one or multiple bare states to higher angular momentum states by the millimeter-wave fields, perhaps through multi-photon transitions. Experiments described here were performed with triplet Rydberg states. Working with the less-dense manifold of singlet states would reduce the chance of spurious couplings.

Another trend not visible in the figures, but that emerges from the fits, is that the parameter characterizing decoherence when probing near the $57s$ state ($\mathcal{C}^{i_{pr}=1}$) in the topological tunneling configuration is much larger, by a factor of 20 or more, than for the other $i_{pr}$, even though the linewidths are not visibly broader. The reason for this is simply that the eigenstate that the probe couples to is almost entirely localized on the edge, with very little weight on other states, so disturbing its superposition with states away from the edge has little effect on the spectra. Therefore, it takes very large $\Gamma_i$ to create any spectral broadening.

Overall, these dependencies of the fit $\Gamma_i$ on $i_{pr}$ may point to deficiencies in the details of the decoherence models. Refining the decoherence models will be interesting for future work. The decoherence is a large contribution to the spectral linewidth for strong millimeter-wave coupling, so identifying the source of this decoherence and eliminating it would greatly expand the types of experiments one could perform with this scheme. There are many reports in the literature of much longer coherence times with Rydberg millimeter-wave spectroscopy[45,46], suggesting this is not an intrinsic limitation. Simulations indicate that for fixed $\Gamma_i$, spectral broadening does not increase with an increasing number of coupled Rydberg states, which is important for increasing the size of the synthetic dimension.

## Discussion

We have demonstrated Rydberg-atom synthetic dimensions as a promising platform for the study of quantum matter. The spectrum of photo-excitation to the synthetic lattice space formed by the manifold of coupled Rydberg levels provides the band structure and decomposition of the lattice eigenstates. SFI of the excited states provides an additional diagnostic of lattice-site populations with two-site resolution. TPS were observed in a six-site SSH model, and the measured band structure and eigenstate decomposition agree well with theory. Varying the detuning of the millimeter-wave fields that create tunneling between sites introduces on-site potentials, and this has been used to break the chiral symmetry of the SSH model and to shift the energies of edge states away from the center of the bandgap at $\Delta_{i_{pr}} = 0$. Numerical simulations based on the master equation were presented, which enable investigation of decoherence effects.

Demonstration of the defining features of the SSH model illustrates the potential of Rydberg-atom synthetic dimensions for quantum simulation. The size of the synthetic space can be expanded by applying more millimeter-wave frequency components, although this will introduce additional complexity such as the need to use multiple local oscillators and horns to cover a wider range of frequencies. The limits imposed by Rydberg-level decoherence, AC stark shifts, and coupling to ancillary levels need further study, but our initial investigations, as well as previous work demonstrating coherent manipulation of Rydberg-level populations (e.g.[45,46]), indicate that these should be technical, rather than fundamental, complications.

Additional millimeter-wave-coupling schemes and tunneling configurations are possible, such as two-photon transitions and transitions with larger changes in principal quantum number. This will enable creation of higher-dimensional synthetic

lattices[1,3,4] and investigation of systems with non-trivial spatial[5] and band-structure[7,8] topologies and higher-order topological states[47], for example. Through control of millimeter-wave phases, tunneling phases around plaquettes and artificial gauge fields can be introduced[48]. This platform is also ideally suited for study of time dependent phenomena, such as Floquet-symmetry-protected states[49], non-equilibrium states[50], and wave-packet dynamics in synthetic space. Tailored time variation of the electric-field ramp[51] may improve site resolution of the SFI diagnostic.

The most exciting prospect is to extend these capabilities to the study of interacting, many-body systems[27,28] using arrays of single Rydberg atoms in closely spaced optical tweezers[52,53] with appreciable long-range dipolar interactions in real space[30] but negligible tunneling of atoms between microtraps. For the Rydberg-level arrangement demonstrated here, the dominant interactions would be flip-flop interactions that couple $|ns, n'p\rangle$ and $|n'p, ns\rangle$ states, giving the many-body Hamiltonian

$$H = -\sum_{i,a} t_i(c_{ia}^\dagger c_{i,a+1} + \text{h.c.}) + \sum_{ij,ab} V_{ij;ab} c_{ib}^\dagger c_{ja}^\dagger c_{ia} c_{jb} \qquad (4)$$

where the $c_{ia}$ and $c_{ia}^\dagger$ are annihilation and creation operators (which can be taken to be either fermionic or bosonic since there is no real-space tunneling) at synthetic site $i$ and real space site $a$, and the interaction matrix elements $V_{ij;ab}$ take the form $V_{ij,ab} = \frac{1 - 3\cos^2\theta_{ab}}{r_{ab}^3} M_{i,j} \delta_{\text{mod}(i-j,2),1}$ where $r_{ab} = |\vec{r}_a - \vec{r}_b|$ is the (real-space) distance between atoms $a$ and $b$, and $\theta_{ab}$ is the angle of $\vec{r}_a - \vec{r}_b$ relative to the quantization axis. Here the quantization axis is the one defining the $m$ levels. The matrix element $M_{i,j}$ falls off rapidly with $|n - n'|^{54}$ and thus $|i - j|$, so, in contrast to many other types of synthetic dimensions, the interactions are highly local in the synthetic space. This can give rise to interesting quantum phases and phase transitions, such as quantum strings and membranes[28,55]. We expect this to be just a small sample of the phenomena these systems can display, with a wide variety of scenarios arising from the easily tunable and dynamic synthetic and real-space geometries.

## Methods

**Experiment**. The laser cooling and trapping of $^{84}$Sr has been described in detail elsewhere[56,57]. Two stages of magneto-optical cooling and trapping are employed corresponding to $5s^2\,{}^1S_0 - 5s5p\,{}^1P_1$ and $5s^2\,{}^1S_0 - 5s5p\,{}^3P_1$ transitions. Atoms are then captured in a 1064 nm crossed-sheet optical dipole trap (ODT), and a short stage of forced evaporation yields samples with $10^5$ atoms, a peak density of about $\sim 10^{11}$ cm$^{-3}$, and a temperature of $T = 2\,\mu$K.

Millimeter-wave frequencies for coupling Rydberg levels are generated by combining outputs of five RF synthesizers (<6 GHz) and mixing the result with a 16 GHz local oscillator. A K-band horn antenna rejects the lower sidebands and directs upper-sidebands to the atoms. The coupling strengths can be varied by varying the low-frequency-synthesizer output powers. Each coupling is calibrated using the Autler-Townes splitting[42] in a two-level configuration.

A 4 Gauss magnetic field splits the $ns$ magnetic sublevels by 11 MHz, which is large compared to tunneling rates. Millimeter waves are resonant or near-resonant with $ns(m = 1) - np$ and $np - (n + 1)s(m = 1)$ transitions for three different $n$'s as shown in Fig. 1b. Millimeter-wave frequencies are adjusted to maintain resonant couplings ($\delta_i \approx 0$) unless disorder is intentionally introduced. AC stark shifts are experimentally determined, and the $\delta_i$ in Eq. (1) are relative to the Stark-shifted Rydberg levels, with uncertainties of 100 kHz for large $J_s$.

Two-photon Rydberg excitation is performed with an intermediate detuning of +80 MHz from the $5s5p\,{}^3P_1$ level[38-40]. For selective field ionization (SFI)[41], an electric field of the form $E(t) = E_p(1 - e^{-t/\tau})$ is applied, with $E_p = 49$ V/cm and $\tau = 6.5\,\mu$s. An atom in level $n\ell$ ionizes at a field given by $\sim 1/[16(n - \alpha_\ell)^4]$, where $\alpha_0 = 3.371$ and $\alpha_1 = 2.887$[58] are the quantum defects of the $ns$ and $np$ states respectively. Liberated electrons are detected by a micro-channel plate, and the Rydberg level, or occupied synthetic-lattice site, can be determined from the arrival time of the electron. Approximately $10^4$ excitation cycles are performed per sample at a 4 kHz repetition rate, and the two-photon drive is weak enough that either zero or one atom is excited to the Rydberg manifold each cycle, even when on a strong resonance peak.

**Theory**. We first discuss effects neglected in the idealized description of the synthetic lattice [Eq. (2)]. We then describe the techniques we use to treat the most important effect, which is decoherence.

There are, in principle, several potential causes of the discrepancies between observations and decoherence-free theory. One is that the idealized analysis has neglected coupling to off-resonant magnetic sublevels of the $ns$ Rydberg states. Theoretical simulations including all these states show they produce level shifts on the order of the observed Stark shifts rather than broadening. Experimentally, we compensate for this shift by measuring it and setting the detuning of the millimeter-waves to the resonant values including the shifts. For the strongest Rabi frequencies we use, the largest shift is about ~500 kHz.

A second effect is the counter-rotating terms that were dropped to arrive at Eq. (2). This effect is negligible since the synthetic tunneling frequencies are at most $J/h \sim 1.5$ MHz, while the frequency difference between any coupled levels is roughly $\Delta E/h \sim 20$ GHz, and therefore effects are expected to be on the order of $10^{-4}$ or smaller. We have also performed numerical simulations of the system that confirm this effect is negligible.

A third possible source of deviations between predicted and observed lineshapes, and likely the most important, is dissipation and decoherence, for which there are multiple potential sources. Spontaneous emission from the Rydberg level or emission stimulated by black-body radiation[41,59] is expected to give a coherence time of >60 $\mu$s for the Rydberg states that we use. This is long compared to the timescale of our experiment. At very low millimeter-wave coupling strength, Rabi spectroscopy for an isolated, two-level 58s-58p system yields full-width-half-maximum linewidths as low as 50 kHz. This is much less than linewidths observed in spectra of synthetic dimensions with couplings $J_s/h \sim 1$ MHz, which implies that fluctuations of stray electric and magnetic fields are not a major source of decoherence for the studies presented in this paper (e.g. Figs. 2 and 4).

Driven 58s-58p Rabi oscillations show that the coherence time decreases with increasing millimeter-wave coupling, with coherence times consistent with values of $\Gamma_i$ derived from theory fits in Figs. 7 and 8 within a factor of 2. This observation motivates our choice of theoretical models for decoherence. Possible sources of decoherence consistent with this observation include coupling to higher angular momentum states and fluctuations in millimeter-wave amplitudes or polarization giving rise to fluctuating AC stark shifts. Further work is required to identify the dominant source of decoherence, and this will be important for determining the ultimate limits on the physics that can be explored.

For theoretical calculations of the spectra, we numerically solve the Lindbladian master equation,

$$\dot{\rho} = -\frac{i}{\hbar}[\hat{H}, \rho] + \sum_{i=1}^{N} \Gamma_i\left[L_i\rho L_i^\dagger - \frac{1}{2}(L_i^\dagger L_i \rho + \rho L_i^\dagger L_i)\right] \qquad (5)$$

where $\rho$ is the density matrix, $\hat{H}$ is the Hamiltonian in Eq. (2), and the jump operators $L_i$ depend on the noise model. For millimeter-wave amplitude noise

$$L_i^{\text{amp}} = \begin{cases} |i\rangle\langle i+1| + |i+1\rangle\langle i| & \text{if } i < N \\ 0 & \text{if } i = N \end{cases}. \qquad (6)$$

The $\Gamma_i$ are determined by fitting spectra, as described in the main text. The noise model extends the corresponding familiar results for 2-level systems. We also performed simulations with a model of white frequency noise and obtained similar results.

In these equations we have included only the magnetic sublevels employed in the synthetic dimension, i.e. those that are resonantly coupled by the millimeter-waves, for notational simplicity. We have also performed theoretical calculations including the off-resonant magnetic sublevels, which are straightforward to include.

## Data availability

Data presented in this publication is available on Figshare with the following identifier. https://doi.org/10.6084/m9.figshare.18258494.

## Code availability

Code for the decoherence simulations and data fitting is available on Figshare with the following identifier. https://doi.org/10.6084/m9.figshare.18259061.

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

## Acknowledgements

This research was supported by the AFOSR FA9550-17-1-0366 (T.C.K.), NSF PHY-1904294 (F.B.D.), NSF PHY-1848304 (K.R.A.H.), and the Robert A. Welch Foundation through grants C-0734 (F.B.D.), C-1844 (T.C.K.), and C-1872 (K.R.A.H.). The authors thank S. Yoshida for helpful conversations.

## Author contributions

T.C.K., K.R.A.H., and F.B.D. planned the research. S.K.K. constructed the millimeter-wave system. S.K.K. performed the measurements, with J.D.W. and Y.L. providing assistance. S.K.K. carried out the data analysis. K.R.A.H., M.Y., and S.D. derived the Hamiltonian and developed the decoherence models and numerically calculated spectra. T.C.K. and F.B.D. supervised the experimental, and K.R.A.H. supervised the theoretical work. All authors discussed the research and contributed to editing the manuscript.

## Competing interests

The authors declare no competing interests.
