## [Peer Review File · Nature Communications]

Realizing topological edge states with Rydberg-atom synthetic dimensionsREVIEWER COMMENTS

Reviewer #1 (Remarks to the Author):

The manuscript demonstrates for the first time the realization of a synthetic lattice with Rydberg states. The synthetic dimension is made of 6 Rydberg states, consecutive s and p states coherently coupled by microwave fields. The selective injection from the ground state to the dressed Rydberg manifold combined with selective field ionization allows for single-site detection in the synthetic dimension. To display the potential of the new platform, the coherent coupling is used to realize the SSH model. The local tunability of the tunneling couplings produced by multiple microwave fields is exploited to realize both the topological and the non-topological phase, and also to induce chiral-symmetry breaking and preserving disorder. The single-site detection in the synthetic dimension is exploited to carefully detect the edge states and probe the topological protection in presence of disorder. The experimental results are compared with the theoretical model both in absence and in presence of decoherence. The first comparison essentially shows that the Rydberg synthetic lattice provides a rather clean and well-calibrated realization of the SSH model. The second comparison is used to determine the main source of decoherence, which seems to be of technical origin and probably related to microwave beams.

I find the work excellent and very interesting. The use of Rydberg states as sites of a synthetic dimension is very promising. The experimental results are very clean and the available size of the synthetic dimension seems to be restricted only by technical limitations. As noticed by the authors, there is the prospect both of including synthetic gauge fields in the synthetic manifold, by replacing the microwave coupling with two-photon transitions, and to include many-body phenomena, for instance by trapping the atoms in optical tweezer arrays. The manuscript is very clearly written and the overview of the existing literature rather complete.

In view of the above comments, I recommend the publication in the present form.

Few additional comments:

- Although rather common the acronym FWHM, Full width at half maximum, should be defined in the text

- I think there is a typo at the line 460, “np- ns(m=1)” should probably be “np-(n+1)s(m=1)”

Reviewer #2 (Remarks to the Author):

The article "Rydberg-atom synthetic dimensions: Realizing topological edge states" by Kanungo et al. describes the spectroscopic investigation of a 6 site Su-Schrieffer-Heeger model in synthetic space. The

novel aspect of this work is the realization using different Rydberg excitation levels in 84Sr as the synthetic dimension. The article is well written and the presented measurements also support the results well. I see the main achievement of this work to be a demonstration of a technique which can be applied to systems of interacting Rydberg atoms in optical tweezer arrays as also stated by the authors in the discussion. These systems have become one of the benchmark tools for quantum simulation with neutral atoms and the synthetic dimensions realized in this work would add to their possibilities. In this respect this work is timely and likely to be very relevant for the field and I therefore principally support publication of the work. However I want to raise one main point which in my opinion weakens the conclusions of this work, and which I would like the authors to address:

The work discusses a relatively simple scenario with 6 lattice sites realized in synthetic space. In this regard this experiment can be seen as a proof of principle demonstration of the technique and of course this is an important step for any new technique. I find however that the authors remain too vague on its limitations. I am convinced that the scenario studied here was chosen making tradeoffs between different experimental parameters. The authors state for instance by additional millimeter wave frequencies 'it should be straightforward to significantly expand the size of the synthetic space'. I understand that this has probably not been done in this work due to technical limitations but I assume there are also limitations or tradeoffs imposed by physics. I feel that this discussion is a bit limited in the manuscript in its current form. I would appreciate if the authors could add a more elaborate discussion on which complications are actually hindering a more complicated system in the current experiment which would help to understand future challenges in using this technique.

Here are some more minor comments I like the authors to address:

- As a non specialist on Rydberg-atom excitation it seems that increasing the 2-photon excitation duration and decreasing the 2-photon laser linewidth would directly improve the spectroscopic resolution. I do not find any discussion on the limitations of doing so in the text. I assume this would also clarify a bit more the limitations of the technique in general.

- From the method section I understand that only zero or one atom of the sample of 10^5 atoms is excited with each laser pulse, correct? For me as a reader it would help to have an idea what the typical excitation probability in the peaks of the observed spectra are, or possibly even replacing the vertical axis in the spectra with detected excitations or probabilities instead of a.u..

Reviewer #3 (Remarks to the Author):

In this work, Kanungo and colleagues report on the experimental realization of a Su-Schrieffer-Heeger chain by using as “lattice sites” different Rydberg levels of Sr atoms, coupled using resonant or quasi resonant microwave fields. This type of approach to quantum simulation, where a spatial dimension is mimicked by internal degrees of freedom, is said to rely on a synthetic dimension and has found increasing interest in the last years.

The paper presents a nice and quite textbook-like example of this approach, showing the main characteristics of the SSH model in a rather minimal chain of six sites. It is relatively easy to read and the careful experimental work is supplemented by significant theoretical modeling. All in all I would recommend publication in Nature Communications once the authors have addressed the points below.

- In the introduction, when citing works using synthetic dimensions to realize topological matter, I think Chalopin et al, Nature Physics 16, 1017-1021(2020) should be cited.
- In its current state, the work is more a minimal proof-of-principle than a useful quantum simulation, for the two following reasons (i) the number of sites is very limited, just 6; (ii) the physics at play is single-particle, that is easy to simulate even with classical resonators for instance, so the real frontier is in adding interactions. This is OK, but I think the authors should develop more their comments on ways to overcome these limits in the future. Concerning point (i), with this Rydberg encoding of the synthetic dimension, in principle one could think of encoding maybe tens of sites, what are the physical or technical reasons why they chose just six? How many sites would realistically be doable? Concerning (ii), the authors write about using dipole-dipole interactions for atoms in tweezer arrays to implement interactions between the particles in synthetic space, I think they should describe the type of interaction Hamiltonian one would obtain.
- Concerning the modeling of decoherence, which seems to have been done quite thoroughly, I’m afraid I do not understand the real conclusion by the authors... Is the decoherence mechanism clarified?
- A minor point, there are numerous typos in the subscript “pr” in $\Delta i_{\{pr\}}$ and the like.
- Another minor wording issue in lines 512-513 “This is negligible on the timescale of our experiment”, I guess saying “much longer” rather than “negligible” would make more sense.

We thank all the reviewers for their careful reading of the manuscript and the many helpful comments. We are pleased that they found the work novel, interesting, and clearly presented. We agree with reviewer 2 that the potential to combine the new techniques presented here with atoms in optical tweezers to study interacting systems is a very exciting direction to pursue.

Detailed responses to each comment are provided below. A revised manuscript is included that highlights all changes.

Reviewer #1:

1.1- “Although rather common the acronym FWHM, Full width at half maximum, should be defined in the text”

We have added the definition.

1.2- “I think there is a typo at the line 460, $n^p - n^s(m=1)$, should probably be $n^p - (n+1)^s(m=1)$ ”

We have made the correction.

Reviewer #2:

2.1 “... this experiment can be seen as a proof of principle ... I find however that the authors remain too vague on its limitations. I am convinced that the scenario studied here was chosen making tradeoffs between different experimental parameters. The authors state for instance by additional millimeter wave frequencies 'it should be straightforward to significantly expand the size of the synthetic space'. I understand that this has probably not been done in this work due to technical limitations but I assume there are also limitations or tradeoffs imposed by physics. I felt that this discussion is a bit limited in the manuscript in its current form. I would appreciate if the authors could add a more elaborate discussion on which complications are actually hindering a more complicated system in the current experiment which would help to understand future challenges in using this technique. “

The decision to work with 6 levels was driven by two main considerations. First, we felt that six levels demonstrated the physics of the SSH model sufficiently. Second, six states connected by five frequencies was a natural break point in the complexity and cost of the millimeter-wave equipment, so it was the target of our initial investment in equipment that was new to our lab. . None of the measurements or calculations we have conducted indicate that we are encountering any limitations that would prevent successful utilization of further microwave couplings when a more complex millimeter-wave system is available.

However, at some point, it is reasonable to expect that with a larger number of frequency components we will encounter accidental degeneracies in transitions to other levels. We are constructing new

Rydberg excitation lasers for creating singlet Rydberg states in order to work with a lower density of states.

It is also possible that at some point, decoherence becomes a serious limitation, but the theory with the decoherence models presented in this paper suggest this is not the case for the types of spectra presented here.

We have changed the statement in the discussion from “By applying more millimeter-wave frequency components, it should be straightforward to significantly expand the size of the synthetic space.” to “The size of the synthetic space can be expanded by applying more millimeter-wave frequency components, although this will introduce additional complexity such as the need to use multiple local oscillators and horns to cover a wider range of frequencies.”

2.2- “As a non specialist on Rydberg-atom excitation it seems that increasing the 2-photon excitation duration and decreasing the 2-photon laser linewidth would directly improve the spectroscopic resolution. I do not find any discussion on the limitations of doing so in the text. I assume this would also clarify a bit more the limitations of the technique in general. “

In general, improving the spectral resolution would be beneficial.

Increasing the pulse duration increases the resolution if other broadening mechanisms are not significant, but this not the case currently in the experiment.

The decoherence discussed in the theory section and included in the fits is already limiting the resolution. This is why we investigated the decoherence and developed theory that could incorporate it. For the models of decoherence we investigated, theory suggests that the resolution does not get worse with increasing number of levels, which is an important point.

Nonetheless, Identifying the source of this decoherence and eliminating it would greatly facilitate the experiments we could perform with this scheme. There are many reports in the literature of much longer coherence times with Rydberg millimeter-wave spectroscopy, so we are confident that it is not a fundamental effect and we will be able to overcome this problem.

The laser linewidth is dominated by the UV photon, which comes from a doubled red laser. The red laser is stabilized to a high-finesse ULE cavity, and presumably has a 50 kHz linewidth. This is respectable, although not state of the art. This is something we would like to improve, but the decoherence appears to be a more critical thing to focus on as the next experimental upgrade.

We have modified the last paragraph before the discussion to read

“The decoherence is a large contribution to the spectral linewidth for strong millimeter-wave coupling, so identifying the source of this decoherence and eliminating it would greatly expand the types of experiments one could perform with this scheme. There are many reports in the literature of much longer coherence times with Rydberg millimeter-wave spectroscopy [45, 46], suggesting this is not an intrinsic limitation.”

2.3- “From the method section I understand that only zero or one atom of the sample of 10^{15} atoms is excited with each laser pulse, correct? For me as a reader it would help to have an idea what the typical

excitation probability in the peaks of the observed spectra are, or possibly even replacing the vertical axis in the spectra with detected excitations or probabilities instead of a.u.. “

The average number of excitations per pulse in the sample is set to an average much less than one for the peak excitation rate. We have clarified that in the manuscript.

“weak enough that either zero or one atom is excited to the Rydberg manifold each cycle, even when on a strong resonance peak.”

The excitation probability per atom per laser shot would be an extremely small number, $\sim 10^{-6}$, which is essentially a bit less than one over the number of atoms. We would prefer not to label the signal strength in this way, and we hope the changed text will provide sufficient clarity on the rate.

Reviewer #3:

3.1- “In the introduction, when citing works using synthetic dimensions to realize topological matter, I think Chalopin et al, Nature Physics 16, 1017-1021(2020) should be cited. “

We have added the reference.

3.2- “In its current state, the work is more a minimal proof-of-principle than a useful quantum simulation, for the two following reasons (i) the number of sites is very limited, just 6; (ii) the physics at play is single-particle, that is easy to simulate even with classical resonators for instance, so the real frontier is in adding interactions. This is OK, but I think the authors should develop more their comments on ways to overcome these limits in the future. “

3.2.1- “Concerning point (i), with this Rydberg encoding of the synthetic dimension, in principle one could think of encoding maybe tens of sites, what are the physical or technical reasons why they chose just six? How many sites would realistically be doable? “

Please see the response to 2.1.

As stated in the conclusion, the limitations due to Decoherence, AC Stark shifts and coupling to ancillary levels need further study. We won't have a rigorous estimate of the maximum number of levels until we push the limits experimentally. Ten or twenty levels seems reasonable. There are still basic questions to explore such as whether $n \sim 60$ is the optimal range in which to work.

A practical consideration is that if one attempts to work with ~ 30 levels, such as $n=60$ to $n=45$ s and p states, in the ladder configuration as used here, the millimeter wave frequencies will vary significantly and will require multiple different millimeter-wave horns and local oscillators. Bulk and optical access will eventually be a limitation.

We added

“The size of the synthetic space can be expanded by applying more millimeter-wave frequency components, although this will introduce additional complexity such as the need to use multiple local oscillators and horns to cover a wider range of frequencies.”

With the scheme presented here, tens of levels are feasible, but not hundreds.

3.2.2- “Concerning (ii), the authors write about using dipole-dipole interactions for atoms in tweezer arrays to implement interactions between the particles in synthetic space, I think they should describe the type of interaction Hamiltonian one would obtain.”

We concur with the Referee’s earlier comment regarding creating interacting systems – which are in our opinion the most exciting direction forward. Specifically, we agree that the present results are a proof of principle rather than direct quantum simulations themselves (at least in the sense of simulating classically intractable phenomena).

To further connect our results to future experiments with atoms in tweezer arrays and lay the groundwork for them, we have added the many-body Hamiltonian as suggested by the referee. For the Rydberg-level arrangement demonstrated here, the dominant interactions would be flip-flop interactions that couple $|ns, n'p\rangle$ and $|n'p, ns\rangle$ states with a matrix element that falls off rapidly with $|n-n'|$. This can qualitatively be seen by calculations with hydrogen atom wavefunctions, and is confirmed in more quantitative calculations (S. Yoshida, personal communication, 2019). We note that although the interactions do have an interesting staggered structure in the synthetic space, where interactions between nearest-neighbor synthetic pairs of sites with the same n (e.g. the pair $i=1,2$) are much stronger than for those with differing n (e.g. the pair with $i=2,3$). We also have added a description of some physics that may occur in these systems, such as quantum strings and membranes. However, we emphasize there is an enormous range of physics possible depending on the synthetic band structure and real-space geometry.

3.3- “Concerning the modeling of decoherence, which seems to have been done quite thoroughly, I’m afraid I do not understand the real conclusion by the authors; Is the decoherence mechanism clarified? “

We state in the section on numerical simulations that “The dominant experimental source of decoherence is not yet determined.”

To make this even more clear, we have added in the paragraph just before the discussion, “The decoherence is a large contribution to the spectral linewidth for strong millimeter-wave coupling, so identifying the source of this decoherence and eliminating it would greatly expand the types of experiments one could perform with this scheme. There are many reports in the literature of much longer coherence times with Rydberg millimeter-wave spectroscopy [45, 46], suggesting this is not an intrinsic limitation.”

3.4- “A minor point, there are numerous typos in the subscript “ p_r ”, $\Delta i_{\{pr\}}$ and the like.”

We have made the corrections.

3.5- “Another minor wording issue in lines 512-513 “This is negligible on the timescale of our experiment”, I guess saying “much longer” rather than “negligible” would make more sense.”

We have changed this to “This is long compared to the timescale of our experiment.”

Regards,

Thomas C. Killian

REVIEWERS' COMMENTS

Reviewer #2 (Remarks to the Author):

I am pleased with the changes made by the author's in the revised manuscript and the reply to my previous comments and support publication.

Reviewer #3 (Remarks to the Author):

The authors have satisfactorily answered my concerns, as well as those of the other referees. I recommend publication of the manuscript.